# Testing the Price of Healthy and Current Diets in Remote Aboriginal Communities to Improve Food Security: Development of the Aboriginal and Torres Strait Islander Healthy Diets ASAP (Australian Standardised Affordability and Pricing) Methods

**DOI:** 10.3390/ijerph15122912

**Published:** 2018-12-19

**Authors:** Amanda Lee, Meron Lewis

**Affiliations:** 1School of Public Health, Faculty of Medicine, The University of Queensland, Herston, Queensland 4006, Australia; m.lewis@uq.edu.au; 2The Australian Prevention Partnership Centre, The Sax Institute, Ultimo 2007, New South Wales, Australia

**Keywords:** food security, diet price, food price, affordability, food policy, nutrition policy, fiscal policy, obesity prevention, non-communicable disease, monitoring and surveillance, INFORMAS

## Abstract

Aboriginal and Torres Strait Islander peoples suffer higher rates of food insecurity and diet-related disease than other Australians. However, assessment of food insecurity in specific population groups is sub-optimal, as in many developed countries. This study tailors the Healthy Diets ASAP (Australian Standardised Affordability and Pricing) methods protocol to be more relevant to Indigenous groups in assessing one important component of food security. The resultant Aboriginal and Torres Strait Islander Healthy Diets ASAP methods were used to assess the price, price differential, and affordability of healthy (recommended) and current (unhealthy) diets in five remote Aboriginal communities. The results show that the tailored approach is more sensitive than the original protocol in revealing the high degree of food insecurity in these communities, where the current diet costs nearly 50% of disposable household income compared to the international benchmark of 30%. Sixty-two percent of the current food budget appears to be spent on discretionary foods and drinks. Aided by community store pricing policies, healthy (recommended) diets are around 20% more affordable than current diets in these communities, but at 38.7% of disposable household income still unaffordable for most households. Further studies in urban communities, and on other socioeconomic, political and commercial determinants of food security in Aboriginal and Torres Strait Islander communities appear warranted. The development of the tailored method provides an example of how national tools can be adapted to better inform policy actions to improve food security and help reduce rates of diet-related chronic disease more equitably in developed countries.

## 1. Introduction

### 1.1. Background

Poor diet is now the major preventable risk factor contributing to the burden of disease globally [1]. In Australia, Aboriginal and Torres Strait Islander peoples suffer the poorest health of all population groups and have a lower life expectancy [2]. At least 75% of the mortality gap between Aboriginal and Torres Strait Islanders and other Australians is attributed to diet-related chronic diseases such as cardiovascular disease, chronic kidney disease, and type 2 diabetes [3]. Malnutrition is a major problem in Aboriginal and Torres Strait Islander communities. This includes both over-nutrition, particularly the consumption of too many ‘discretionary’ food and drinks (those not necessary for health, that are high in saturated fat, added sugar, salt and/or alcohol), and under-nutrition, particularly dietary deficiencies related to inadequate intake of healthy foods in the five food groups and unsaturated spreads and oils allowance, as recommended by the Australian Dietary Guidelines [3,4]. Forty-one percent of the energy intake reported by Aboriginal and Torres Strait Islanders in the Australian Health Survey (AHS) 2011–2013 was derived from ‘discretionary’ food and drinks [5]. This was higher than reported by non-Indigenous Australians, for whom 35% of the energy intake of adults and 39% of the energy intake of children was derived from discretionary choices [6]. 

Few Australians (<4%) consume diets consistent with the Australian Dietary Guidelines (ADGs) [4,7]. The contribution of poor diets to the rising rates of overweight and obesity associated with chronic disease is of particular concern. Twenty-five percent of all Australian children aged two to 17 years and 63% of Australian adults aged 18 years and over are overweight or obese [8]. These proportions are even higher for Aboriginal and Torres Strait Islander groups, with 30% of children aged two to 17 years and 66% of adults being overweight or obese [3]. Nutrition policy actions are needed urgently to improve the current diet of the whole Australian population and particularly of Aboriginal and Torres Strait Islander groups. 

Good nutrition is underscored by food security. This is when “all people, at all times, have physical, social and economic access to sufficient, safe and nutritious food that meets their dietary needs and food preferences for an active and healthy life” [9]. Food security has been deemed to be a fundamental human right [10]. The Universal Declaration of Human Rights affirms that “everyone has the right to a standard of living adequate for the health and well-being of himself and of his family, including food” [11]. The right to adequate food has been seen as “a right of people to be given a fair opportunity to feed themselves, now and in the future” [12] rather than a right to be fed. In this way, food security is impacted by the availability, accessibility, affordability and acceptability (appropriateness) of the food supply. The experience of these determinants of food security can vary greatly amongst different groups of the population in developed economies like Australia [3,4].

One specific barrier to food security in Australia is believed to be the relative expense of healthy foods. This is particularly the case, among low socioeconomic groups [13,14,15,16,17] in which Aboriginal and Torres Strait Islanders are over-represented. More than one in five Aboriginal and Torres Strait Islanders reported living in a household that had run out of food in the past year and had not been able to afford to buy more in 2011–2013 [18]. This proportion was much higher than in the non-Indigenous population (3.7%) [6]. The affordability of healthy food is believed to be a key aspect of the inequitable distribution of household food security in developed economies such as Australia [13] and a major challenge to food security in remote Aboriginal and Torres Strait Islander communities particularly [3,19]. For over twenty years food prices have been shown consistently to be around 30% higher in remote Aboriginal and Torres Strait Islander communities than in urban centres [20], yet median household incomes are lower in remote areas than in urban areas [3].

However, past food price surveys in Australia have applied a wide variety of ‘food basket’ costing tools and methods [20] and results are not comparable across different locations or times due to dissimilarity of metrics in the different approaches [20]. These include: number and type of foods surveyed; application of availability and/or quality measures; definition of reference households; estimated household income calculation methods; food store sampling frameworks; data collection methods; and analysis [20]. Until recently, standardised methods to assess and compare the price and affordability of healthy diets with currently consumed, unhealthy diets were lacking in Australia [20] and globally [21]. Such methods are essential to provide robust, meaningful data to inform health and fiscal policy actions, for example, decisions around exemption of basic, healthy foods from Goods and Services Tax (GST) and introduction of health levies on sugary drinks [17,22].

The Healthy Diets ASAP methods protocol was developed to assess, compare and monitor the price and affordability of healthy and current diets among the general population in Australia [22,23]. The method was based on the ‘optimal’ approach to monitor food price and affordability globally proposed by the International Network for Food and Obesity/non-communicable Diseases Research, Monitoring and Action Support (INFORMAS) [21]. Surprisingly, testing of the Healthy Diets ASAP methods protocol demonstrated that the price of healthy diets recommended by the Australian Dietary Guidelines were 12–15% less expensive than reported current (unhealthy) diets in Australia [22,24]. The results also suggested that Australians were spending 49–64% of their household food budget on discretionary foods and drinks [22,24].

In addition to application at international and national levels, the ‘optimal’ approach of the INFORMAS diet price and affordability framework [21] has the potential to be modified for use in specific populations and localities. This allows for comparison of diet price and affordability in specific population groups and locations with that of the general population, to inform the development of targeted health and fiscal policies. For example, in Australia, the Healthy Diets ASAP approach has been applied in country Victoria, as reported elsewhere in this special edition [25]. As another example, the INFORMAS ‘optimal’ approach to assessing diet price and affordability has been tailored to different population groups in New Zealand [26]; results showed that a healthy diet would be more affordable than the current diet for both the total New Zealand population (3.5% difference) and Pacific households (4.5% difference) but the cost of both diets would be similar for Māori households (0.57% difference). However, while previous surveys have used market baskets to estimate the price of ‘healthy’ basic foods in remote Aboriginal and Torres Strait Island communities [3,20,27], the ‘optimal’ approach in the INFORMAS step-wise framework to monitor food price data [20] had not been adapted for use in Aboriginal and Torres Strait Islander groups in Australia to enable generation of policy-relevant data.

### 1.2. The Healthy Diets ASAP Methods Protocol 

The Healthy Diets ASAP methods protocol for application with the Australian population as a whole has been reported in detail elsewhere [23]. The protocol consists of five parts; (i) construction of the healthy (recommended) and current (unhealthy) diet pricing tools, (ii) calculation of both median and low-income household incomes; (iii) store location and sampling, (iv) price data collection, and (v) analysis and reporting. To modify the protocol for Aboriginal and Torres Strait Islander groups only the first and second parts of the protocol required adjustment. The remaining three parts of the protocol were retained exactly to optimise comparability of results.

Part one of the Healthy Diets ASAP methods protocol covers the development of two diet pricing survey tools. These are the current (unhealthy) diet pricing tool and the healthy (recommended) diet pricing tool. The current (unhealthy) diet pricing tool comprises the mean fortnightly intake of specific foods and drinks reported in the AHS 2011–2013, expressed in grams or millilitres, by each age/gender group corresponding to the four individuals comprising a reference household (an adult male 31–50 years old, an adult female 31–50 years old, a 14-year-old boy and an eight-year-old girl) in the AHS 2011–2013 [28]. The amounts of foods and drinks consumed per day were derived from the AHS 2011–2013 Confidentialised Unit Record Files (CURFs) of reported dietary intake at 5-digit code level [28]. The mean reported daily dietary intakes for the four individuals were multiplied by 14 to produce the quantities consumed per household per fortnight. The healthy (recommended) diet pricing tool reflects the types and amounts of corresponding foods and drinks for the reference household for a fortnight consistent with the ADGs [4]. In both diet pricing tools, an allowance for edible portion foods/as cooked, as specified in AUSNUT 2011-13 [28], was included; however, post-plate wastage was not estimated or included.

In the second part of the Healthy Diets ASAP methods protocol pertaining to household income, median household income is sourced from national Australian census data which provide a total (gross) amount per household per week (i.e. before taxation). To estimate household income at time points between the five-yearly census, national wage price indexes (published quarterly) are applied [29].

The indicative low (minimum) income for the household is calculated from minimum wage and welfare payments provided by the Department of Human Services [30,31]. A set of assumptions relating to employment, housing type, education attendance, disability status, savings and investments and children’s immunisation status are used to determine the appropriate welfare payments and taxation payable. As taxation payable is included, the indicative low (minimum) income is considered disposable income.

Affordability of the healthy and current diets for the reference household is determined by comparing the cost of each diet with the median (gross) household income and with the indicative low (minimum) disposable income of low income households per fortnight. Internationally, a benchmark of 30% of income has been used as a cut-off point to indicate affordability of a diet [16,21].

### 1.3. Aim

The aim of this study was to modify and test the Healthy Diets ASAP methods protocol to be more relevant to the Aboriginal and Torres Strait Islander population. It developed methods and tools to assist others to apply the approach in order to compare the price, price differential and affordability of healthy (recommended) and current (unhealthy) diets of Aboriginal and Torres Strait Islanders living in different locations with other population groups in Australia.

## 2. Methods

### 2.1. Development of the Aboriginal and Torres Strait Islander Healthy Diets ASAP Methods 

It was not necessary to amend the Healthy Diets ASAP healthy (recommended) diet pricing tool to adapt the Healthy Diets ASAP methods protocol for application with Aboriginal and Torres Strait Islander groups as the Australian Dietary Guidelines already include culturally-appropriate and commonly available food and drink options and are similar at broad food group level for both Aboriginal and Torres Strait Islanders and non-Indigenous people [4].

However, the Healthy Diets ASAP current (unhealthy) diet pricing tool required modification to reflect the mean intake of each relevant Aboriginal and Torres Strait Islander age and gender group in the National Aboriginal and Torres Strait Islander Nutrition and Physical Activity Survey component of the AHS 2011–2013 [5]. This was compared with the mean dietary intake of each relevant age and gender group of the whole Australian population reported in the AHS 2011–2013 [5,6] to calculate a reported consumption ratio for each food group or, where data were available, for component food and drinks in each food group. This ratio was applied to derive estimates of the current dietary intake of all foods and drinks included in the current diet pricing tool in Aboriginal and Torres Strait Islander groups. 

In relation to assessment of household income, it was not necessary to adapt the Healthy Diets ASAP methods protocol to determine the median (gross) household income in Aboriginal and Torres Strait Islander communities in remote areas as census data is reported for relevant Statistical Areas (SA2). 

However, assumptions regarding characteristics of the household members were reviewed in relation to any welfare and taxation policies specific to Aboriginal and Torres Strait Islander people and/or to those people living in remote locations in order to better reflect Aboriginal and Torres Strait Islander households living in remote areas for the calculation of indicative low (minimum) disposable income [30,32]. The current quantums of relevant welfare and taxation payments were applied to calculate the indicative low (minimum) disposable household income.

### 2.2. Testing of the Aboriginal and Torres Strait Islander Healthy Diets ASAP Methods

Prices of food and drinks were collected in five community stores on the Anangu Pitjantjatjara Yankunytjatjara (APY) Lands of South Australia (Figure 1) using the Healthy Diets ASAP food price data collection sheet as per the Healthy Diets ASAP methods protocol [23] by AL in June 2017 as part of ongoing Nganampa Health Council service delivery. In each location, a single store is the main source of food in the community. Further information about the communities is available elsewhere [27].

Under the Healthy Diets ASAP pricing collection methods protocol, several discretionary food prices are collected from commercial premises outside of a supermarket, including for pizza, hamburger, beef pie and hot chips. As such premises were not available in the remote communities, relevant prices were collected from the store’s hot takeaway section, or if not available, for frozen pizza, frozen hamburger and frozen pies, as these items were commonly heated by the purchaser in a microwave at the store for immediate consumption. Frozen potato chip prices were not collected however, due to the large price differential between a single serve of hot chips and the significantly larger bag of frozen chips, and the requirement for more complex cooking methods than microwaving. Alcohol prices were not collected as the communities are ‘dry’ and alcohol was not available for sale.

Price data were double entered by ML into data entry and analysis Excel© spreadsheets developed for the Healthy Diets ASAP methods protocol [23,33]. If the price of a specific food item was unavailable, the average price for that item in the other four stores was used. If an item was out of stock, the shelf price was collected. The mean prices for each diet and component food groups were calculated. The data were analysed according to both the Healthy Diets ASAP methods protocol and the Aboriginal and Torres Strait Islander Healthy Diets ASAP methods, and results were compared.

Median (gross) household income data from the Community Profile for the APY Lands SA2 [34] was transcribed directly and adjusted by the Wage Price Index percentage increase from June 2016 (at census data collection) to June 2017 (when the food price data were collected) [29]. 

### 2.3. Ethical Standards Disclosure

The QUT University Human Research Ethics Committee assessed this study as meeting the conditions for exemption from Human Research Ethics Committee review and approval in accordance with section 5.1.22 of the National Statement on Ethical Conduct in Human Research (2007); the exemption numbers are 1500000161 and 1800000151. All data were obtained from publicly available sources and did not involve human participants.

## 3. Results

### 3.1. Developing the Aboriginal and Torres Strait Healthy Diets ASAP Methods Part One: Construct of the Diet Pricing Tools

#### 3.1.1. The Aboriginal and Torres Strait Islander Healthy Diets ASAP Current (Unhealthy) Diet Pricing Tool

The reported consumption ratio calculated by comparing the reported dietary intakes of each food group, and relevant components, by Aboriginal and Torres Strait Islanders [5] with the whole Australian population [7] in the AHS 2011–2013 is presented in Table 1.

The reported consumption ratios were applied to calculate the amounts of foods and drinks comprising the Aboriginal and Torres Strait Islander Healthy Diets ASAP current (unhealthy) diet pricing tool as presented in Table 2. The composition of the original Healthy Diets ASAP current (unhealthy) diet pricing tool for the whole population is also presented in Table 2.

#### 3.1.2. The Aboriginal and Torres Strait Islander Healthy Diets ASAP Healthy (Recommended) Diet Pricing Tool

The amounts of foods and drinks comprising the Aboriginal and Torres Strait Islander Healthy Diets ASAP healthy (recommended) diet pricing tool (unchanged from the original protocol [23]) are presented in Table 2. 

### 3.2. Developing the Aboriginal and Torres Strait Healthy Diets ASAP Methods Part Two: Determination of Median and Low-Income Household Income

The Community Profile for the APY Lands SA2 states that the median weekly household income in 2016 was $AUD1150 and the average household contained 3.8 people [34]. The Australian Wage Price Index [29] increased from June 2016 (at census data collection) to June 2017 (when the food price data was collected) by 1.9%. Applying this index gave an estimated median weekly household income on the APY Lands in June 2017 of $AUD1171. Thus, the fortnightly median (gross) household income for the reference household in the APY Lands in June 2017 was $AUD2342.

The method to determine the indicative low (minimum) disposable household income was modified to include in the underlying assumptions (Table 3) that the reference family is comprised of people identifying as Aboriginal and/or Torres Strait Islanders and that they live in a remote location as determined by the Australian Tax Office [32]. Compared with the Healthy Diets ASAP methods protocol, the assumptions included an AbStudy school term allowance for the 14-year-old boy [30] and a remote area tax offset amount applied in assessment of taxation for the adult male, as shown in Table 4. All other assumptions were the same as those for non-Indigenous households and those living in non-remote areas [30].

### 3.3. Testing the Aboriginal and Torres Strait Islander Healthy Diets ASAP Methods: The Cost of The Diets and Component Food Groups

The mean (± standard deviation) cost of the healthy diet was $AUD827.63 (± $42.24) in the five remote Aboriginal communities surveyed.

The cost of the current diet in the five remote Aboriginal communities using the Healthy Diets ASAP methods protocol and Aboriginal and Torres Strait Islander Healthy Diets ASAP methods, and the difference between the two, are presented in Table 5. Application of the Aboriginal and Torres Strait Islander Healthy Diets ASAP methods assessed the cost of the current diet at $AUD1023.16 (± $40.90) which was 7% higher than the cost of $AUD956.18 (± $39.60) assessed by application of the original Healthy Diets ASAP methods protocol.

Using the Aboriginal and Torres Strait Islander Healthy Diets ASAP methods, the cost of a healthy diet was 24% less than the cost of the current diet. If the original Healthy Diets ASAP methods protocol was used, the cost of the healthy diet was 16% less than the current diet.

Using the Healthy Diets ASAP methods protocol, the proportion of the total cost of the current diet derived from discretionary foods and drinks was 56.5%. This figure was 62.0% when the Aboriginal and Torres Strait Islander Healthy Diets ASAP methods were used.

The total cost of the current (unhealthy) diet was $AUD66.97 per fortnight (7%) more expensive when the Aboriginal and Torres Strait Islander specific methods were used rather than the original Healthy Diets ASAP methods protocol for the whole population (Table 5). The main source of difference for healthy foods was that the cost of all fruit and vegetables included in the current diet was $AUD25.37 per fortnight (19%) less when assessed by the Aboriginal and Torres Strait Islander Healthy Diets ASAP methods than by the Healthy Diet ASAP methods protocol. The healthy unsaturated oils and spreads were cost 30% less using the Aboriginal and Torres Strait Islander specific methods; however, this was a difference of only $AUD0.61 per fortnight, given the low quantities of these foods consumed. Conversely, the cost of all unhealthy discretionary foods and drinks included in the current diet was $AUD94.37 (17%) more expensive per fortnight when assessed by the Aboriginal and Torres Strait Islander Healthy Diets ASAP methods than by the Healthy Diets ASAP methods protocol. The major source of this variance was the cost of sugar-sweetened drinks which were $AUD47.18 per fortnight (69%) more expensive when assessed by the Aboriginal and Torres Strait Islander specific methods. 

### 3.4. Testing of the Aboriginal and Torres Strait Islander Healthy Diets ASAP Methods: Affordability of Healthy Diets

The affordability of healthy diet and the current diets determined by both the Healthy Diets ASAP methods protocol and the Aboriginal and Torres Strait Islander Healthy Diets ASAP methods in five remote Aboriginal communities are shown in Table 6. When determined by the Aboriginal and Torres Strait Islander Healthy Diets ASAP methods, the affordability of the current diet was around 7% poorer than when assessed by the original Healthy Diets ASAP methods protocol. When assessed by the Aboriginal and Torres Strait Islander Healthy Diets ASAP methods, the affordability of the current diet as a proportion of both the median (gross) household income (35.3%) and indicative low (minimum) disposable household income (38.7%) respectively was above the internationally acceptable benchmark of 30% [16,21]. As assessed by the Aboriginal and Torres Strait Islander Healthy Diets ASAP methods, the healthy diet would be around 20% more affordable than the current diet, but at 35.3% of the median (gross) household income and 38.7% of the indicative low (minimum) disposable household income, would still be unaffordable compared to the internally acceptable benchmark of 30% [16,21].

## 4. Discussion

### 4.1. Discussion of Approach

Food insecurity is a key factor contributing to the high double-burden of malnutrition experienced by Indigenous Australians [3]. However, as in many developed nations, food security is poorly assessed in Australia, where, for over twenty years irregular national dietary surveys have included a single question on individual food security around running out of food and not being able to afford to buy more [3]. This measure, while a useful indicator, is likely to underestimate the full extent of the problem. There is a pressing need to better understand food insecurity from an Aboriginal and Torres Strait Islander perspective in order to develop Indigenous-specific tools for assessment of availability, affordability, accessibility and acceptability of healthy food and drinks and other determinants of food security, particularly at household and community level [3,35]. This paper attempted to do this in the area of food price and affordability, in order to provide relevant data to inform the development of tailored fiscal and nutrition policy actions with Aboriginal and Torres Strait Islander communities.

Adjustment of the whole-of-population Healthy Diets ASAP current (unhealthy) diet pricing tool by the reported consumption ratio method, proved to be a simple, expedient method to customise the tool for application in remote Aboriginal and Torres Strait Islander communities, particularly as it did not require redevelopment of the original data collection tools. However, this method does rely on the availability of quality dietary (food and drink) intake survey data for both the whole population at the national level and for specific population groups, which may not always be available, even in developed countries [21].

The total cost of the current diet was 7% more expensive when the Aboriginal and Torres Strait Islander specific methods were used rather than the original Healthy Diets ASAP methods protocol for the whole population. This was due to differences in the reported intakes of foods and drinks that contributed substantially to the current diet in Aboriginal and Torres Strait Islander groups compared with the broader Australian population. Major differences were seen for sugar-sweetened drinks (with reported intakes nearly double that of broader Australia) contributing most (69%) of the additional expense, and reported intakes of fruit and vegetables (which were 30% less than the broader population) reducing the current diet costs by 19%, when determined by the Aboriginal and Torres Strait Islander Healthy Diets ASAP methods.

While data on median (gross) household income of Aboriginal and Torres Strait Islanders specifically are not readily available, the use of median (gross) household income from the relevant 2016 Census data Community Profiles [34] did provide meaningful information once updated with the wage price index [29], and at $AUD2342 per household per fortnight (gross), was consistent with expectations given the low (minimum) disposable household income of $AUD2139 estimated in the test communities using different methods. This study demonstrated that determination of the indicative low (minimum) disposable household income for Aboriginal and Torres Strait Islander households living in remote areas was feasible. This figure for Aboriginal and Torre Strait Islander households living in non-remote areas would be slightly less, due to the non-applicability of the remote area tax offset.

Testing demonstrated that it is feasible to apply the Aboriginal and Torres Strait Islander Healthy Diets ASAP methods stores in remote communities. However, further studies would be required to test utility of the approach in urban centres. Among other differences, remote community stores, stock a much smaller range of items than supermarkets in urban areas. In this study there were four instances where the listed food item in the pricing tool was unavailable in any size or brand. Each store outlet surveyed in the five remote communities operates as a general store selling fresh fruit, vegetables, meat, bread, frozen foods, pantry items, and other goods. Four of the stores also sold a range of hot takeaway food items. Some food items were available only in sizes much smaller than stated on the price collection data sheet; for example, plain yoghurt is listed as 1kg on the Healthy Diets ASAP food price data collection sheet, but was only available in 200g tubs in three of the five stores. This contributed to the high standard deviation in the cost of the food groups observed where larger items were missing, particularly the milk, yoghurt and cheese food group. Conversely, as part of the nutrition policy in place in the five stores surveyed, the price of 600mL bottled water is mandated at $AUD1.00, so that for this item the standard deviation of prices across the five stores was zero.

Testing of the Aboriginal and Torres Strait Islander Healthy Diets ASAP methods supported the notion that the approach has acceptable face validity in providing assessment of the price, price differential and affordability of current (unhealthy) and healthy (recommended) diets of Aboriginal and Torres Strait Islander groups living in remote communities. The results were consistent with expectations arising from consideration of the reported dietary intake data of the two different populations [5,7] and the relative prices of foods in the remote Aboriginal community stores [27].

Consistent with similar surveys, particularly in Australia where the GST of 10% is not applied to basic, healthy foods [22,23,24,25], application of the Aboriginal and Torres Strait Islander Healthy Diets ASAP methods in remote Aboriginal communities showed that a healthy diet ($AUD827.63) was less expensive than the current diet ($AUD1,023.16) per household per fortnight. However, at 76% of the cost of the current diet, a healthy diet was potentially more affordable in the remote Aboriginal communities studied than in other places, where the cost of the healthy diet ranges between 80–85% [22,23,24,25]. Surprisingly, this price differential between the current and healthy diets was larger than in other studies even though alcohol was not included in the current diet, as the communities are ‘dry’ and alcohol is not available for sale. One likely reason for this is that the five community stores surveyed on the APY Lands have in place a prescribed nutrition policy which mandates, among other potential benefits, that fruit and vegetables are sold at cost price, that 600mL bottled water is priced at $AUD1.00, and that low mark ups on the wholesale price of other healthy foods, such as lean meat and wholemeal bread, are standard. Previous studies have found that this nutrition policy contributes to relative affordability of healthy foods, particularly fruit and vegetables, compared to unhealthy, discretionary choices [27].

Despite these promising findings, further scrutiny showed that healthy diets would be unaffordable due to the low household incomes in the communities surveyed. When assessed by the Aboriginal and Torres Strait Islander Healthy Diets ASAP methods, healthy diets would cost over 35% of median (gross) household income and nearly 39% of indicative low (minimum) disposable household incomes in these communities, compared to the international affordability benchmark of 30% of disposable household income [16,21]. 

The high level of food insecurity and food stress in these communities was confirmed, as the current diet cost over 43% of median (gross) household income and nearly 48% of indicative low (minimum) disposable household incomes when assessed by the Aboriginal and Torres Strait Islander Healthy Diets ASAP methods.

The tailored Aboriginal and Torres Strait Islander Healthy Diets ASAP methods were more sensitive than the original Healthy Diets ASAP methods protocol in revealing the current degree of food security in the communities surveyed. If the tailored methods developed and tested in this study had not been used, the severity of food security issues in the remote Aboriginal communities surveyed would have been partially masked, and valuable data relevant to potential policy actions would have remained undetected.

Worryingly, while 41% of the energy intake of the diet was derived from discretionary choices [5], application of the Aboriginal and Torres Strait Islander Healthy Diets ASAP methods showed that 62% of the current food budget in the remote communities surveyed was spent on discretionary food and drinks; of this over 18% was spent on sugary drinks and over 30% on take-away foods. This high reliance on discretionary food and drinks has been described previously and appears to be driven by the increasing availability, range and variety of unhealthy discretionary foods and drinks in community stores over the last three decades [27]. Such changes in the food supply reflect those seen more broadly in Australia, and globally [27].

These results highlight that, given the high proportion of food insecurity and diet-related disease in Aboriginal and Torres Strait Islander groups, nothing should be done to risk increasing the price differential of healthy to discretionary food and drinks in remote Aboriginal communities, as this could act as a further barrier to healthy diets. While better understanding of price elasticities and access to income entitlements in remote communities would be useful, the findings also suggest that investigation into the nature and effect of drivers of food choice other than price, such as housing, access to educational and employment opportunities, availability and functionality of food preparation/cooking facilities, transport, convenience, product placement in stores, promotion, advertising and food preferences appears warranted.

### 4.2. Limitations

Similar to the original Healthy Diets ASAP methods protocol, there are several inherent limitations in the Aboriginal and Torres Strait Islander Healthy Diets ASAP methods. Given that the approach is based on the reported mean dietary intakes of select age and gender groups of Aboriginal and Torres Strait Islanders at the national level, the diet pricing tools should be considered as reference instruments and the cost of the current diet is unlikely to be the same as actual expenditure on food and drinks by all Aboriginal and Torres Strait Islander people or households currently [36].

All diet pricing tools should ideally include foods that are culturally acceptable, commonly consumed and widely available. Whilst the amounts of the foods included in the diet survey pricing tools are reflective of the respective food and food group consumption of Aboriginal and Torres Strait Islander groups reported in the AHS 2011-13 [5] at the three digit-level, the Healthy Diet ASAP methods protocol includes foods and drinks reported in the AHS 2011-13 [6] at the five-digit level. Therefore, a very small number of the specific foods and drinks included tend to reflect reported consumption of the Australian population as a whole, rather than Aboriginal and Torres Strait Islander peoples specifically. While all foods in the pricing tools were generally available and accessible in the remote community stores surveyed, formal assessment of their cultural acceptability has not been undertaken as yet. Subsequent modifications may be required to accommodate specific food preferences; for example, further reduction of the quantities of plain yoghurt included as this item was frequently out of stock and was considered by store managers to be a low demand item.

No adjustments were made to account for the marked under-reporting in the AHS 2011-13 [5,6]. Nor were adjustments made for the greater proportion of ‘convenience’ items in the current (unhealthy) diet pricing tool compared with the healthy (recommended) diet pricing tool. Given the high rates of overweight/obesity in Aboriginal and Torres Strait Islander groups, and that the Foundation Diets of the modelling used to inform the Australian Guide to Healthy Eating component of the Australian Dietary Guidelines were prescribed for the shortest and least active in each age group [37], the healthy (recommended) diet tool under-estimates the requirements of taller, more active and healthy weight individuals. 

## 5. Conclusions

The Aboriginal and Torres Strait Islander Healthy Diets ASAP methods tailor nationally-standardised diet price and affordability method protocols to improve applicability to Indigenous Australians. The method incorporates relevant household income data and reported dietary intakes of Aboriginal and Torres Strait Islander groups to more appropriately assess, compare, monitor and benchmark the price, price differential and affordability of current (unhealthy) and healthy (recommended) diets in different communities. 

The development of the tailored Aboriginal and Torres Strait Islander Healthy Diets ASAP methods provides an example of how standardised national tools can be adapted at sub-population and regional levels to provide better data to inform policy actions to improve food security and help reduce rates of diet-related disease more equitably in developed countries.

## Figures and Tables

**Figure 1 ijerph-15-02912-f001:**
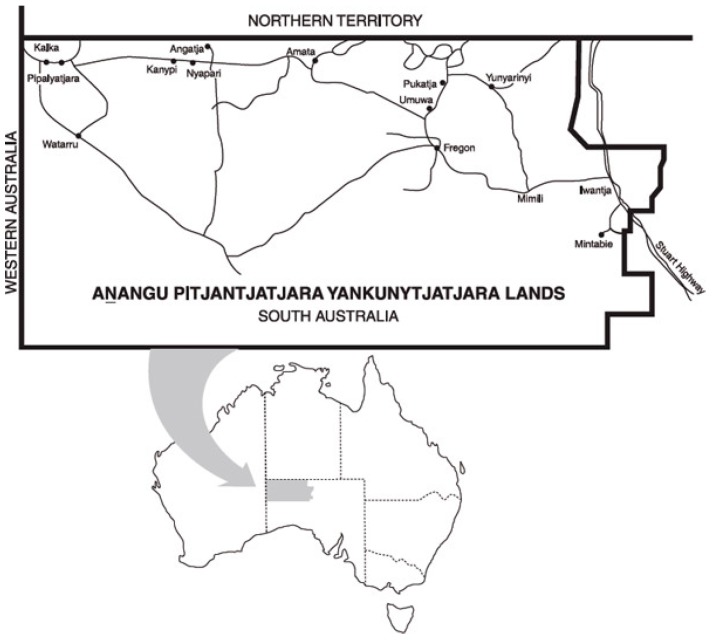
Map showing the Anangu Pitjantjatjara Yankunytjatjara (APY) Lands of South Australia.

**Table 1 ijerph-15-02912-t001:** Reported consumption ratios of each food group and food group component for Aboriginal and Torres Strait Islanders [5] compared with the broader Australian population [7].

Food Group and Food Group Component	Reported Consumption Ratio
Vegetables & legumes	0.67
Fruit	0.8
Grain (cereal) foods—wholegrains	0.74
Grain (cereal) foods—others	1
Meat, poultry, fish & alternatives—red meat & poultry	1.14
Meat, poultry, fish & alternatives—others	0.94
Milk, yoghurt, cheese & alternatives	0.8
Unsaturated oils & spreads	0.7
Discretionary foods—sugar-sweetened drinks	1.8
Discretionary foods—others	1.1
Water	1
Alcohol	1

**Table 2 ijerph-15-02912-t002:** Composition of the Healthy Diets ASAP and Aboriginal and Torres Strait Islander Healthy Diets ASAP current (unhealthy) diet and healthy (recommended) diet pricing tools (pertaining to the diet of a reference household per fortnight ^1^).

Current (Unhealthy) Diet Pricing Tool			Healthy (Recommended) Diet Pricing Tool	
Food or Drink	Healthy Diets ASAP Quantity	Aboriginal and Torres Strait Islander Healthy Diets ASAP Quantity	Food or Drink	Healthy Diets ASAP and Aboriginal and Torres Strait Islander Healthy Diets ASAP Quantity
Bottled water, still (mL)	5296	5296	Bottled water, still (mL)	5296
Artificially sweetened ‘diet’ soft drink (mL)	2391	2630		
*Fruit*			*Fruit*	
Apples, red, loose (g)	3497	2797	Apples, red, loose (g)	5460
Bananas, Cavendish, loose (g)	899	719	Bananas, Cavendish, loose (g)	5460
Oranges, loose (g)	1664	1331	Oranges, loose (g)	5460
Fruit salad, canned in juice (g)	2046	1637		
Fruit juice	3026	3329		
*Vegetables*			*Vegetables*	
Potato, white, loose (g)	1460	978	Potato, white, loose (g)	2320
Sweetcorn, canned, no added salt (g)	206	138	Sweetcorn, canned, no added salt (g)	1160
Broccoli, loose (g)	422	282	Broccoli, loose (g)	1470
White cabbage, loose (g)	235	157	White cabbage, loose (g)	1470
Iceberg lettuce, whole (g)	795	533	Iceberg lettuce, whole (g)	1470
Carrot, loose (g)	753	505	Carrot, loose (g)	2205
Pumpkin (g)	240	161	Pumpkin (g)	2205
Four bean mix, canned (g)	74	50	Four bean mix, canned (g)	1005
Diced tomatoes, canned, in tomato juice (g)	234	157	Diced tomatoes, canned, in tomato juice (g)	1638
Onion, brown, loose (g)	84	57	Onion, brown, loose (g)	1638
Tomatoes, loose (g)	488	327	Tomatoes, loose (g)	1638
Frozen mixed vegetables, pre-packaged (g)	1184	793	Frozen mixed vegetables, pre-packaged (g)	1638
Frozen peas, pre-packaged (g)	273	183	Frozen peas, pre-packaged (g)	1638
Baked beans, canned (g)	369	247	Baked beans, canned (g)	1005
Salad vegs in sandwich	120	120	Salad vegs in sandwich	120
Veg in tinned meat and vegetable casserole (g)	646	736		
*Grain (cereal) foods*			*Grain (cereal) foods*	
Wholegrain cereal biscuits Weet-bix^TM^ (g)	430	319	Wholegrain cereal biscuits Weet-bix^TM^ (g)	2216
Wholemeal bread, pre-packaged (g)	1054	780	Wholemeal bread, pre-packaged (g)	4272
Rolled oats, whole (g)	870	870	Rolled oats, whole (g)	6648
White bread, pre-packaged (g)	3033	3033	White bread, pre-packaged (g)	893
Cornflakes (g)	680	680	Cornflakes (g)	670
White pasta, spaghetti (g)	1326	1326	White pasta, spaghetti (g)	2042
White rice, medium grain (g)	1622	1622	White rice, medium grain (g)	2042
Dry water cracker biscuit (g)	258	258	Dry water cracker biscuit (g)	781
Bread in sandwich	120	120	Bread in sandwich	120
*Meats, poultry, fish, eggs, nuts and seeds*			*Meats, poultry, fish, eggs, nuts and seeds*	
Beef mince, lean (g)	267	305	Beef mince, lean (g)	1168
Lamb loin chops (g)	257	293	Lamb loin chops (g)	1169
Beef rump steak (g)	1056	1204	Beef rump steak (g)	1172
Tuna, canned in vegetable oil (g)	1052	989	Tuna, canned in vegetable oil (g)	1841
Whole barbeque chicken, cooked (g)	1661	1893	Whole barbeque chicken, cooked (g)	1471
Eggs (g)	872	820	Eggs (g)	2208
Meat in tinned meat and vegetable casserole (g)	646	736	Peanuts, roasted, unsalted (g)	780
Chicken in sandwiches	120	120	Chicken in sandwiches	120
*Milk, yoghurt, cheese and alternatives*			*Milk, yoghurt, cheese and alternatives*	
Cheddar cheese, full fat (g)	624	499	Cheddar cheese, full fat (g)	704
Cheddar cheese, reduced fat (g)	44	35	Cheddar cheese, reduced fat (g)	516
Milk, full fat (mL)	5961	4769	Milk, full cream (mL)	6438
Milk, reduced fat (mL)	2929	2344	Milk, reduced fat (mL)	12000
Yoghurt, full fat plain (g)	204	163	Yoghurt, full fat plain (g)	2576
Yoghurt, reduced fat, flavoured (vanilla) (g)	676	541	Yoghurt, reduced fat, flavoured (vanilla) (g)	5100
Flavoured milk (mL)	2416	2658		
*Unsaturated oils*				
Canola margarine (g)	170	119	Canola margarine (g)	412
Sunflower oil (mL)	7	5	Sunflower oil (mL)	291
Olive oil (mL)	7	5	Olive oil (mL)	291
*Discretionary choices*				
Beer, full strength (mL)	4661	4661		
White wine, sparkling (mL)	863	863		
Whisky (mL)	266	266		
Red wine (mL)	1078	1078		
Butter (g)	280	308		
Muffin, commercial (g)	1455	1601		
Cream-filled sweet biscuit, pre-packaged (g)	496	546		
Muesli bar, pre-packaged (g)	373	410		
Mixed nuts, salted (g)	255	281		
Pizza, commercial (g)	1182	1300		
Savoury flavoured biscuits (g)	222	244		
Confectionary (g)	418	460		
Chocolate (g)	441	485		
Sugar-sweetened beverages (Coca Cola) (mL)	12012	21621		
Meat pie, commercial (g)	1638	1802		
Frozen lasagne, pre-packaged (g)	4322	4754		
Hamburger, commercial (g)	2413	2654		
Beef sausages (g)	1048	1152		
Ham (g)	189	208		
Potato crisps, pre-packaged (g)	518	570		
Potato chips, hot, commercial (g)	670	737		
Ice cream (g)	1830	2013		
White sugar (g)	564	621		
Salad dressing (mL)	277	305		
Tomato sauce (mL)	569	626		
Chicken soup, canned (g)	1340	1474		
Orange juice (mL)	3027	3330		
Fish fillet crumbed, pre-packaged (g)	302	332		
Instant noodles, wheat-based (g)	381	419		

^1^ The reference household comprises four people: adult male 19–50 years old; adult female 19–50 years old; boy 14 years old; girl 8 years old.

**Table 3 ijerph-15-02912-t003:** Assumptions used to determine the indicative low (minimum) disposable household income of the reference household.

Assumptions for the Reference Household Consisting of an Adult Male, an Adult Female, a 14-Year-Old Boy and an 8-Year-Old Girl
The family is privately renting a house at $AUD75/week [34] The adult male works on a permanent basis at the national minimum wage ($AUD17.29 per hour [31]) for 38 h a week
The adult female works on a part-time basis at the national minimum wage ($AUD17.29 per hour) for 6 h a week
Both children attend school and are fully immunised
None of the family are disabled
The family has some emergency savings that earn negligible interest
The family are Aboriginal and/or Torres Strait Islanders The family live in a remote location

**Table 4 ijerph-15-02912-t004:** Calculation of the indicative low (minimum) disposable household income of the reference Aboriginal and Torres Strait Islander household.

Income Type	Amount Per Fortnight ($AUD)
Paid employment–adult male	1345.20
Paid employment–adult female	212.40
Family Tax Benefit A	420.70
Family Tax Benefit A Supplement	55.87
Family Tax Benefit B	108.64
Family Tax Benefit B Supplement	13.62
Clean Energy Supplement (across all payments)	9.94
Rent Assistance	nil
AbStudy School term allowance–14 yr old boy	20.80
Income tax paid (tax owing on employment income of adult male, less low income tax offset, less remote area tax offset	−48.09
**Total Fortnightly Income**	**2139.08**

**Table 5 ijerph-15-02912-t005:** Mean cost of the current (unhealthy) diet in five remote Aboriginal communities using the Healthy Diets Australian Standardised Affordability and Pricing (ASAP) methods protocol and Aboriginal and Torres Strait Islander Healthy Diets ASAP methods.

	Healthy Diets ASAP (Whole Population) Methods Protocol		Aboriginal and Torres Strait Islander Healthy Diets ASAP Methods		Cost Difference between Using Aboriginal and Torres Strait Islander Healthy Diets ASAP Methods and Healthy Diets ASAP Methods Protocol
Diet Component	Mean Cost ($AUD)	Std Dev ($AUD)	Mean Cost ($AUD)	Std Dev ($AUD)	Difference $AUD (% Change)
Water	8.83	---	8.83	---	---
Fruit	80.66	8.70	68.13	6.93	−12.54 (−15%)
Vegetables & Legumes	56.31	3.21	43.48	2.15	−12.83 (−23%)
Grains & Cereals	66.42	2.35	63.37	2.29	−3.05 (−5%)
Meats, nuts, seeds, eggs	110.65	3.33	119.95	3.16	9.30 (+8%)
Milk, yoghurt, cheese	80.66	13.72	71.93	14.08	−8.73 (−11%)
Unsaturated oils & spreads	2.03	0.01	1.42	0.01	−0.61 (−30%)
Artificially sweetened soft drink	10.52	---	11.57	---	1.05 (+10%)
Take-away foods	181.32	17.53	199.45	19.28	18.13 (+10%)
Sugar-sweeteneddrinks	68.18	---	115.35	---	47.18 (+69%)
Discretionary choices-other	290.62	13.36	319.68	14.70	29.06 (+10%)
**Total cost**	**956.18**	**39.60**	**1023.16**	**40.90**	**66.97 (+7%)**

^1^ The current diet for a fortnight for the reference household comprising four people: adult male 19–50 years old; adult female 19–50 years old; boy 14 years old; girl 8 years old.

**Table 6 ijerph-15-02912-t006:** Affordability of current diets and healthy diets in remote Aboriginal communities on the APY Lands.

Diet	Mean Diet Cost (±Std Dev) ($AUD)	Affordability with Median (Gross) Household Income ($AUD2342)	Affordability with Indicative Low (Minimum) Disposable Household Income ($AUD2139.08)
Healthy (recommended) diet	827.63 (42.24)	35.3%	38.7%
Current (unhealthy) diet determined by the Healthy Diets ASAP methods protocol	956.18 (39.60)	40.8%	44.7%
Current (unhealthy) diet determined by the Aboriginal and Torres Strait Islander Healthy Diets ASAP methods	1023.16 (40.90)	43.7%	47.8%

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
