# Peer review of "Testing the Price of Healthy and Current Diets in Remote Aboriginal Communities to Improve Food Security: Development of the Aboriginal and Torres Strait Islander Healthy Diets ASAP (Australian Standardised Affordability and Pricing) Methods"

_ijerph, 2018, doi:10.3390/ijerph15122912_

Round 1
Reviewer 1 Report
This is a routinely written paper following standard scientific methods to establish an appropriate tool to measure the cost of diet in the population of Aborigines and Torres Strait Islanders. Though my comments may look lengthy, they are by nature only of minor dimension:
Title: I think that the title is too long. Should be shortened to “Testing the Price of Healthy and Current Diets in remote Aboriginal Communities to improve food security: Development of the Aboriginal and Torres Strait Islander Healthy Diets ASAP (Australian Standardised Affordability and Pricing) Methods” (which is still quite a long text string for a title)
l. 21-27. Difficult to follow all implications in this sentence. Please split it up.
l. 21-27 (and rest of the text): The terms “price” and “cost” are used interchangeably. I think that it would be more appropriate to use the term “price” in connotation with foods and “cost” for what the individuals/households have to spend for their diets. Hence, in l. 22, it rather should say “cost” and not “price”.
l. 22, 31,41,45… The rather long term “Aboriginal and Torres Strait Islander Healthy Diets” is used repeatedly. I suggest to using an abbreviation (e.g. ATSIHD) to facilitate easy reading.
l. 48 The authors should add a definition (and evtl. characterisation) of what they consider to be “healthy foods”. At this place, I rather think that the term “micronutrients” would be more appropriate.
l. 80: Replace “Australian Standardised Affordability and Pricing (ASAP)” by ASAP.
l. 103 Unclear where the “optimal” claim comes from. How is this justified?
l. 116: The repeated usage of the terms “current (unhealthy)” and healthy (recommended)” seems redundant. With sufficient justification at one point indicating why the consumption of the current diet is associated with an increased health risk and why the recommended diet is health-promoting, one could use the terms “current” and “recommended”, respectively, only.
l. 197: Remove the redundant period.
l. 240: The presentation of one of the right columns seems redundant as they are identical. So having one column with the value represented twice with the combined header “Quantity of the Healthy Diets ASAP and Aboriginal and Torres Strait Islander Healthy Diets ASAP” would do the job better.
l. 241. I’m not happy with the composition of the reference household as I consider it to be biased towards the male part of the population. Particularly in face of the higher risk of women to develop iron deficiency-based anaemia and the intake of iron-providing foods, the food amounts should have been calculated for a reference household comprising two adults, a boy aged 8 yrs and a girl aged 14 yrs.
l. 272-296 The text section describing Table 5 should emphasize the crucial role in the cost estimation difference of the two tools for soft drinks; the overall 7% difference depends on the 47.2% difference for soft drink cost.
l. 357-359: This sentence is difficult to understand – please reformulate.
l. 358. It’s unclear to me why the authors use the term “differential” and not “difference”.
In the Discussion, I miss the comment on the extra-ordinal impact of the cost difference on soft drinks. Without the +69% cost difference, the calculated overall +7% difference would be void, the latter serving as an important element of the entire Discussion.
Author Response
Point one: This is a routinely written paper following standard scientific methods to establish an appropriate tool to measure the cost of diet in the population of Aborigines and Torres Strait Islanders. Though my comments may look lengthy, they are by nature only of minor dimension:
Response one: Thank you for your considered and constructive feedback, which has helped us improve the paper.
Point two: Title: I think that the title is too long. Should be shortened to “Testing the Price of Healthy and Current Diets in remote Aboriginal Communities to improve food security: Development of the Aboriginal and Torres Strait Islander Healthy Diets ASAP (Australian Standardised Affordability and Pricing) Methods” (which is still quite a long text string for a title)
Response two: Thank you. The title has been shortened as proposed.
Point three: l. 21-27. Difficult to follow all implications in this sentence. Please split it up.
Response three: Thank you. The sentence has been rewritten in two shorter sentences.
Point four: l. 21-27 (and rest of the text): The terms “price” and “cost” are used interchangeably. I think that it would be more appropriate to use the term “price” in connotation with foods and “cost” for what the individuals/households have to spend for their diets. Hence, in l. 22, it rather should say “cost” and not “price”.
Response: Agreed. This matches our intentions and all terms have been checked and replaced where necessary- in total eight uses of the term ‘price’ were replaced with ‘cost’ where the latter reflected what households would need to spend for their diets. Thank you for picking up.
Point five: l. 22, 31,41,45… The rather long term “Aboriginal and Torres Strait Islander Healthy Diets” is used repeatedly. I suggest to using an abbreviation (e.g. ATSIHD) to facilitate easy reading.
Response five: Aboriginal and Torres Strait Islander Peoples and Organisations in Australia have expressed strong objection to the abbreviation of these cultural population descriptors terms, and the use of 'ATSI' is considered offensive by Aboriginal and Torres Strait Islander Peoples who feel the term disrespects cultural diversity in Australia. We agree that that the preferred terminology appears long, but will not risk offending our colleagues by inappropriate shortening. We would not advise publishing the paper if such abbreviations are to be used. Therefore we feel unable to respond to this request on cultural grounds.
Point six: l. 48 The authors should add a definition (and evtl. characterisation) of what they consider to be “healthy foods”. At this place, I rather think that the term “micronutrients” would be more appropriate.
Response six: The source of the definition of ‘healthy’ foods and diets in this paper is as defined and recommended by the Australian National Health and Medical Research Council’s Australian Dietary Guidelines, as per the included citations (reference 4); these healthy foods fall into the five food groups and unsaturated spreads and oils allowance in these guidelines (reference 4). We have clarified these points within the manuscript. Although nutrient data are available in the modelling document that informed the development of the Australian Dietary Guidelines (available at www.eatforhealth.gov.au) food-based approach to improving nutrition and food security is culturally-preferred as more meaningful by Aboriginal and Torre Strait Islander groups than a (micro) nutrient-based approach, which may be additional justification for the food-based definition of ‘healthy’ foods included in the paper. We are also keen to use a nationally accepted policy tool, such as the Australian Dietary Guidelines to define terms, in order to increase uptake and utility of the approach.
Point seven: l. 80: Replace “Australian Standardised Affordability and Pricing (ASAP)” by ASAP.
Response seven: Agreed. Completed as requested.
Point eight: l. 103 Unclear where the “optimal” claim comes from. How is this justified?
Response eight: The term ‘optimal’ approach is a direct quote from the INFORMAS step-wise framework to monitor food price data presented in reference 21 (revised numbering). This has now been explained more clearly within the manuscript and the reference citation has been moved to be positioned closely with this explanation.
Point nine: l. 116: The repeated usage of the terms “current (unhealthy)” and healthy (recommended)” seems redundant. With sufficient justification at one point indicating why the consumption of the current diet is associated with an increased health risk and why the recommended diet is health-promoting, one could use the terms “current” and “recommended”, respectively, only.
Response nine: Agreed. The expanded terms have been used in this manuscript given these are the terms in the Healthy Diets ASAP methods protocol paper; this came about in response to a reviewer of that protocol paper, who had requested consistent use of the expanded terms throughout that paper prior to publication. However, with the exception of cases where the dietary pricing tools in the protocol are being referred to directly or where emphasis is potentially useful, we have shortened the terms in this manuscript as requested.
Point ten: l. 197: Remove the redundant period.
Response ten: Completed.
Point eleven: l. 240: The presentation of one of the right columns seems redundant as they are identical. So having one column with the value represented twice with the combined header “Quantity of the Healthy Diets ASAP and Aboriginal and Torres Strait Islander Healthy Diets ASAP” would do the job better.
Response eleven: Thank you. The format of contents and header of Table 2 have been amended as requested.
Point twelve: l. 241. I’m not happy with the composition of the reference household as I consider it to be biased towards the male part of the population. Particularly in face of the higher risk of women to develop iron deficiency-based anaemia and the intake of iron-providing foods, the food amounts should have been calculated for a reference household comprising two adults, a boy aged 8 yrs and a girl aged 14 yrs.
Response twelve: The composition of the reference household was determined, along with other decisions requiring expert opinion, by stakeholder consensus at the National Food Price and Affordability Summit held in March 2016, as outlined in the cited Healthy Diets ASAP methods protocol paper (reference 23- revised numbering). Consideration by participants included composition of households used previously in longitudinal food price data sets, Australian demographic data and specific nutritional needs through the lifespan. As with all tools, the most important issue is that they are used, so having consensus amongst policy, practice, academic and community stakeholders was a key issue. As the ASAP method/analysis involves comparison of the cost of two diets, it is arguably less important what reference family is used; this point is illustrated by the similar results for the other four household compositions tested, as described in reference 22 (revised numbering).
Point thirteen. l. 272-296 The text section describing Table 5 should emphasize the crucial role in the cost estimation difference of the two tools for soft drinks; the overall 7% difference depends on the 47.2% difference for soft drink cost.
Response thirteen: Thank you. This is an important point and it is agreed that it should have been emphasised better. Additional text clarifying and emphasising this point has been added to the paragraph as suggested. In addition the term ‘soft drink’ has been changed to ‘sugar sweetened drink’ throughout the manuscript for greater specificity. (In addition, as per response to point sixteen, this point has been more clearly emphasised in the Discussion too).
Point fourteen: l. 357-359: This sentence is difficult to understand – please reformulate.
Response thirteen: Thank you. The sentence, and surrounding context, have been rephrased for greater clarity.
Point fifteen: l. 358. It’s unclear to me why the authors use the term “differential” and not “difference”.
Response fifteen: It is understood that the term ‘difference’ and ‘differential’ are synonyms. The latter was used in the original INFORMAS paper (reference 21- revised numbering) and in the Healthy Diet ASAP methods protocol paper (reference 23- revised numbering) on the advice of respected international health economists, and we have retained this variation in this current manuscript as we do not wish to deviate from the methods protocol without strong reason.
Point sixteen: In the Discussion, I miss the comment on the extra-ordinal impact of the cost difference on soft drinks. Without the +69% cost difference, the calculated overall +7% difference would be void, the latter serving as an important element of the entire Discussion.
Response sixteen: Thank you. An additional paragraph covering this point has now been added to the Discussion.
Reviewer 2 Report
The association between diet-related mortality and the percentage of discretionary food expenditures presented in the lines 48 to 52 are not convincing. How could a 6% difference in discretionary expenditures between Aboriginals and non-indigeneous Australians explain 75% of the mortality gap? Because food prices will be higher in the northern territory due to logistical reasons and most probably (this information is actually MISSING) the overall food budget of non-indegeneous Australians is higher, the purchase quantity of discretionary foods are most probably very similar if not higher than for aborigenes. It is the quantity of “overnutrition” that could be a health issue not the share of food budget.
Furthermore there some inconsistent share of expenditures in the text. Line 50 is written 35% on line 88 49-64%. What share now?
On the different estimates of the costs of healthy and unhealthy diet. Sorry I am confused as many estimates are used throughout the text. But in a nutshell healthy diets are actually cheaper than unhealthy? Correct? This is the opposite of the findings in the USA (Drewnowski). Drewnoski actually postulates that the lower costs of unhealthy diet in the main driver of consumers choices. In Australia it would be different? Unhealthy diet as a kind of superior good to healthy diet? It does not make sense honestly. I am also confused with the statement that healthy diets would be unaffordable. Well if unhealthy are more expensive how can they be affordable? In the sentence 392 the authors mention nutrition policy contributes to relative affordability of healthy foods. So what would be the conclusion? even more price controls on healthy food? As fare as i remember, shop owner in remote area have no real economic interest in having lots of fruits and vegetables because they are perishibale and make no money so they are often simply out of stock. Again, there is intervention trial amongst the aborigenes population that showed that price subsidy on fruits and vegetables and tax on soda led actually to increase of SSB consumption because of the “wealth effects” of the lower prices on healthy products will increase the budget available for unhealthy diet. So really I am lost.
The main objection is however: what is the main objective of the paper? That the adapted method better reflects the reality of expenditures and uncover underestimates? Than it should be better highlighted. Mixing this main point with “health policy promotion does not really help and some association between food budget share and diet related mortality is not helping.
Author Response
Point one: The association between diet-related mortality and the percentage of discretionary food expenditures presented in the lines 48 to 52 are not convincing. How could a 6% difference in discretionary expenditures between Aboriginals and non-indigeneous Australians explain 75% of the mortality gap?
Response one: The figures presented in lines 42 to 48 are derived accurately from a recent review of authoritative sources (reference 3), but appear to have been misinterpreted by this reviewer. It is not claimed in the manuscript that 6% difference in discretionary expenditure explains 75% of the mortality gap as claimed in point one. The text states that “at least 75% of the mortality gap between Aboriginal and Torres Strait Islanders and other Australians is attributed to diet-related chronic diseases such as cardiovascular disease, chronic kidney disease and type 2 diabetes [3].” It goes on to state that “Malnutrition is a major problem in Aboriginal and Torres Strait Islander communities. This includes both over-nutrition, particularly the consumption of too many ‘discretionary’ food and drinks (those not necessary for health, that are high in saturated fat, added sugar, salt and/or alcohol), and under-nutrition, particularly dietary deficiencies related to inadequate intake of healthy foods,” hence clearly identifying the broad dietary determinants of the diet-related diseases that contribute to this unacceptable mortality gap.
Further, as one example component of the nature of the dietary differences between Aboriginal and Torres Strait Islander Australians and non-Indigenous Australians, the text presents clearly the data for contribution of ‘discretionary foods and drinks’ to energy intake in both population groups, not difference in discretionary expenditure, as claimed by this reviewer. The current text is “Forty-one percent of the energy intake reported by Aboriginal and Torres Strait Islanders in the Australian Health Survey (AHS) 2011-13 was derived from ‘discretionary’ food and drinks [5]. This was higher than reported by non-Indigenous Australians, for whom 35% of the energy intake of adults and 39% of the energy intake of children was derived from discretionary choices [6]”. This point is made clearly.
Point two: Because food prices will be higher in the northern territory due to logistical reasons and most probably (this information is actually MISSING) the overall food budget of non-indegeneous Australians is higher, the purchase quantity of discretionary foods are most probably very similar if not higher than for aborigenes. It is the quantity of “overnutrition” that could be a health issue not the share of food budget.
Response two: The information included in the Introduction (lines 70-74) about the relative unaffordability of foods in remote Aboriginal communities and challenges with measuring this has been expanded considerably to provide greater detail and clarity around both cost of food and household income in remote areas. As framed in the Introduction, the purpose of this manuscript, as outlined in section 1.3 Aim, is to test assumptions similar to those made by the reviewer, using total diet as an ‘anchor’ to enable comparison of dietary intake and food and diet expenditure in the different population groups, as presented in the Results and elaborated in the Discussion.
Point three: Furthermore there some inconsistent share of expenditures in the text. Line 50 is written 35% on line 88 49-64%. What share now?
Response three: 35% doesn’t appear on line 50, but the test in the relevant areas is “Forty-one percent of the energy intake reported by Aboriginal and Torres Strait Islanders in the Australian Health Survey (AHS) 2011-13 was derived from ‘discretionary’ food and drinks [5]. This was higher than reported by non-Indigenous Australians, for whom 35% of the energy intake of adults and 39% of the energy intake of children was derived from discretionary choices [6].” In the latter sentence the 35% refers clearly to the proportion of energy intake that is derived from discretionary foods and drinks as reported by Australian adults in the national nutrition survey.
Line 88 reports “The results also suggested that Australians were spending 49-64% of their household food budget on discretionary foods and drinks.” In this sentence 49-64% refers clearly to the proportion of the household food budget spent on discretionary foods and drinks.
Both these metrics- proportion of dietary energy intake and proportion of food budget spent- are referring to very different measures. Hence the point the reviewer is making is not clear, nor is the intent of the reviewer’s question “What share now?” However, no changes to the text appear warranted.
Point four: On the different estimates of the costs of healthy and unhealthy diet. Sorry I am confused as many estimates are used throughout the text. But in a nutshell healthy diets are actually cheaper than unhealthy? Correct? This is the opposite of the findings in the USA (Drewnowski). Drewnoski actually postulates that the lower costs of unhealthy diet in the main driver of consumers choices. In Australia it would be different? Unhealthy diet as a kind of superior good to healthy diet? It does not make sense honestly.
Response four: As explained in the Introduction (original lines 80-89) relatively recent, systematic enquiry of food and diet pricing methods globally by the INFORMAS group (reference 21-revised numbering) including in Australia (reference 22 and 24- revised numbering ) showed that healthy diets can be less expensive than unhealthy diets: “Surprisingly, testing of the Healthy Diets ASAP methods protocol demonstrated that the price of healthy diets recommended by the Australian Dietary Guidelines were 12-15% less expensive than reported current (unhealthy) diets in Australia.” The referenced citations include several methodological challenges with previous approaches, such as the effect of pricing ‘unanchored’ lists of arbitrarily selected ‘healthy’ and ‘unhealthy’ foods and the spurious coupling affecting the assessment of the cost (per kJ) of foods by their energy density ($/kJ). Further, as explained in the manuscript (original line 419), Australia does not incur 10% GST on basic healthy foods which does assist the relative cost of the healthy diet, although similar results have been found in other countries including New Zealand where 15% GST does apply to basic healthy foods (for example see reference 26). Although it is acknowledged in the manuscript that the relative affordability of healthy diets (as opposed to foods) is surprising, and such findings have been confirmed by this study, this is not the main aim of this manuscript (see response to point six below). Conversely this issue has been covered in great detail in the referenced citations (reference 21, 22, 24 and 26 –revised numbering).
Point five: I am also confused with the statement that healthy diets would be unaffordable. Well if unhealthy are more expensive how can they be affordable?
Response five: As highlighted in the methods and the limitations, the current diet pricing tool is based on the national mean reported dietary intake of Aboriginal and Torres Strait Islander individuals comprising the reference household, and the cost of this is “unlikely to be the same as the actual current expenditure on food and drinks by all Aboriginal people or household” including those on the APY Lands. As noted in the Discussion (original lines 395-400) the fact that it would cost more than the international benchmark of 30% of household income to purchase this diet is an indication of food stress in these communities: “The high level of food insecurity and food stress in these communities was confirmed, as the current diet cost over 43% of median (gross) household income and nearly 48% of indicative low (minimum) disposable household incomes when assessed by the Aboriginal and Torres Strait Islander Healthy Diets ASAP methods.”
The mean national healthy diet would be more affordable, but still not affordable in these communities “When assessed by the Aboriginal and Torres Strait Islander Healthy Diets ASAP methods, healthy (recommended) diets would cost over 35% of median (gross) household income and nearly 39% of indicative low (minimum) disposable household incomes in these communities, compared to the international affordability benchmark of 30% of disposable household income.” This again indicates the level of food stress and food insecurity impacting on these communities.
It’s not quite clear what the reviewer is missing from the comments made, but a statement clarifying that the diet pricing tools are used as ”reference instruments” has been added to the limitations in case this is useful.
Point six: In the sentence 392 the authors mention nutrition policy contributes to relative affordability of healthy foods. So what would be the conclusion? even more price controls on healthy food? As fare as i remember, shop owner in remote area have no real economic interest in having lots of fruits and vegetables because they are perishibale and make no money so they are often simply out of stock. Again, there is intervention trial amongst the aborigenes population that showed that price subsidy on fruits and vegetables and tax on soda led actually to increase of SSB consumption because of the “wealth effects” of the lower prices on healthy products will increase the budget available for unhealthy diet. So really I am lost.
Response six: The solutions to Aboriginal and Torres Strait Islander nutrition and food security are complex and beyond the scope of this targeted paper; however, as noted in the Conclusions, this manuscript “provides an example of how standardized national tools can be adapted at sub-population and regional levels to provide better data to inform policy actions to improve food security and help reduce rates of diet-related disease more equitably in developed countries.” Nutrition issues and strategies to improve Aboriginal and Torres Strait Islander food security and diet-related health, including the study mentioned in point six, have been reviewed comprehensively and summarised recently in the cited reference three.
Point seven: The main objection is however: what is the main objective of the paper? That the adapted method better reflects the reality of expenditures and uncover underestimates? Than it should be better highlighted. Mixing this main point with “health policy promotion does not really help and some association between food budget share and diet related mortality is not helping.
Response seven: As per section 1.3. “The aim of this study was to modify and test the Healthy Diets ASAP methods protocol to be more relevant to the Aboriginal and Torres Strait Islander population. It developed methods and tools to assist others to apply the approach in order to compare the price, price differential and affordability of healthy (recommended) and current (unhealthy) diets of Aboriginal and Torres Strait Islanders living in different locations with other population groups in Australia” so is similar to the suggested objective, but is designed to better reflect dietary “intakes” to develop food pricing tools, not expenditures. The potential expenditures are outcome measures, not inputs. Interpretation of the data was assisted by knowledge of the existence of the local food pricing policies in the stores surveyed. The adjusted approach has potential to inform both fiscal and health policies at various levels, and we believe it is important to mention these in Discussion as they are highly relevant to the utility of the methods developed. The intent of the point raised about the association between food budget share and diet related mortality and desired response is unclear, as such an association is not claimed in the manuscript. As per response to point one and point three, above it poor diet underscored by food insecurity and malnutrition (in all forms) that is associated with diet-related mortality.
Reviewer 3 Report
This is an interesting and well-performed study of food security issues in remote aboriginal communities in Australia. I have only one recommendation on the possible improvement of the manuscript - to address the pillars of food security, particularly, their implementation to the specific situation in aboriginal communities. The author provides a general definition of food security (lines 63-65). In my opinion, to understand various dimensions of food security situation in the territories under investigation, the discussion of food security has to be expanded (various approaches to understanding food security, four pillars, measurements, etc.)
Author Response
Point one: This is an interesting and well-performed study of food security issues in remote aboriginal communities in Australia.
Response One: Thank you
Point two: I have only one recommendation on the possible improvement of the manuscript - to address the pillars of food security, particularly, their implementation to the specific situation in aboriginal communities. The author provides a general definition of food security (lines 63-65). In my opinion, to understand various dimensions of food security situation in the territories under investigation, the discussion of food security has to be expanded (various approaches to understanding food security, four pillars, measurements, etc.)
Response two: Agreed. The section on food security has been expanded in the Introduction as suggested. This provided a better framework for the relevant points included in the first paragraph of the Discussion around the “pressing need to better understand food insecurity from an Aboriginal and Torres Strait Islander perspective in order to develop Indigenous-specific tools for assessment of availability, affordability, accessibility and acceptability of healthy food and drinks and other determinants of food security, particularly at household and community level.”
Thank you for your constructive feedback which has helped improve the manuscript.